# Effectiveness of Psychological First Aid e-Orientation among the General Population in Muntinlupa, the Philippines

**DOI:** 10.3390/ijerph20020983

**Published:** 2023-01-05

**Authors:** Satoshi Iiyama, Takashi Izutsu, Yuki Miyamoto, John Russel Manuel Benavidez, Atsuro Tsutsumi

**Affiliations:** 1Graduate School of Medicine, The University of Tokyo, Tokyo 113-0033, Japan; 2Graduate School of Arts and Sciences, The University of Tokyo, Tokyo 153-0041, Japan; 3Persons with Disability Affairs Office, City Government of Muntinlupa, Muntinlupa City 1781, Philippines; 4Institute of Transdisciplinary Sciences for Innovation, Kanazawa University, Kanazawa 920-1192, Japan

**Keywords:** Psychological First Aid (PFA), e-learning, capacity building, mental health and psychosocial support, non-specialist

## Abstract

This present study examined the effectiveness of the Psychological First Aid (PFA) e-orientation as well as face-to-face PFA orientation among the general population in Muntinlupa City, the Philippines. The e-orientation group consisted of 150 participants who received a two-hour PFA e-orientation (male: 47, female: 97, others: 6, mean age: 33.4 (SD = 12.1)), the face-to-face (F2F) group consisted of 139 participants who received a two-hour face-to-face PFA orientation (male: 41, female: 95, others: 3, mean age: 35.0 (SD = 13.8)), and the control group consisted of 117 participants who received a two-hour face-to-face health promotion orientation for obesity (male: 48, female: 65, others: 4, mean age: 34.2 (SD = 13.8)). In order to see the effect of these interventions, the confidence to provide PFAs was compared between the pre- and post-interventions in each group with paired t-tests. Further, the number of correct answers regarding the knowledge on PFA was also compared between the pre- and post-interventions utilizing a McNemar test. The results demonstrated that the mean scores on the confidence increased significantly in the e-orientation (pre: 25.1 (SD = 4.7), post: 26.1 (SD = 5.3), *p* = 0.02) and F2F (pre: 26.2 (SD = 6.0), post: 29.6 (SD = 6.9), *p* < 0.01) groups. Regarding knowledge on PFA, in the e-orientation group, the number of those who answered correctly increased significantly in a question (pre: 10, post: 24, *p* = 0.01), and there was a trend for improvement in another question (pre: 63, post: 76, *p* = 0.06). In the F2F group, the number of those who answered correctly increased significantly in two questions (pre: 21, post: 38, *p* < 0.01, and pre: 5, post: 14, *p* = 0.05), and there were trends for improvement in two questions (pre: 69, post: 82, *p* = 0.06, and pre: 17, post: 27, *p* = 0.09), while in the control group, there were no significant differences in any of the questions between pre- and post-intervention. The results suggest that both the PFA e-orientation and face-to-face orientation are effective for the general population in terms of increasing confidence and knowledge related to PFA.

## 1. Introduction

The promotion of mental health and well-being has become increasingly important in recent years and has been included as a key global priority in the United Nations (UN) 2030 Agenda for Sustainable Development and its Sustainable Development Goals (SDGs) [1]. However, financial and human resources for mental health are extremely limited in developing countries, and there is a significant gap between the needs and access to mental health and psychosocial support. For example, the World Health Organization (WHO) reports that the annual government expenditure for mental health per capita is USD 0.02 in low-income countries, USD 1.05 in lower–middle-income countries, USD 2.02 in upper–middle-income countries, and USD 80.24 in high-income countries [2]. Human resources for mental health are also limited: The number of human resources for mental health per 100,000 individuals are 1.6, 6.2, 20.6, and 71.7 in low, lower–middle, upper–middle, and high-income counties, respectively [2]. 

Mental health and well-being are often adversely affected by disasters [3]. The number and complexity of disasters are increasing globally, and many people suffer from them. According to statistics from the UN Office for Disaster Risk Reduction (UNDRR) [4], many enormous disasters have been reported around the world: Between 2000 to 2019 alone, these disasters resulted in economic damages exceeding USD 2.97 trillion while affecting 4.03 billion people, with 1.23 million lives lost [4]. In Asia, known as the most disaster-prone region in the world, 3068 disasters have occurred in the past 20 years, which is greater than 40% of the global total. Additionally, the entire world is currently facing an unprecedented crisis due to the COVID-19 pandemic [4].

According to the Inter-Agency Standing Committee (IASC) Guidelines on Mental Health and Psychosocial Support in Emergency Settings [3], meeting basic needs and providing social support by the community and family members as well as non-professionals and professionals are considered critical to protect and promote mental health and well-being in emergencies. The Psychological First Aid (PFA) Guide for Field Workers [5] was developed as a result of collaboration among UN agencies and international non-governmental organizations (NGOs) as a key component of achieving them. PFA involves “humane, supportive and practical help to fellow human beings suffering from crisis events” [5]. PFA is based on the principle of “do no harm” and is not limited to professionals only. PFA has been actively implemented with a focus on enhancing prevention, response, recovery, and resilience in disaster-prone regions, including the Asia–Pacific region. Figure 1 shows the key components of PFA indicated in the table of contents of the PFA Guide [5].

However, resources such as the opportunity to learn how to offer PFA and find trained instructors are limited, especially in developing countries [2]. It is particularly difficult to receive face-to-face orientation in countries such as the Philippines, which is an archipelagic nation consisting of more than several thousand islands. With the COVID-19 pandemic, the importance of e-learning is recognized rapidly around the world. It is critical to provide opportunities to learn about promoting mental health and psychosocial support, such as PFA, in an accessible way to all.

According to statistics from the UNDRR [4], in terms of the number of disasters that have occurred in the world in the past 20 years, the Philippines ranks fourth, with a total of 304 disasters. There are many types of disaster experiences in the country, including storms, floods, earthquakes, volcanic activities, and fires. However, the WHO reports that the total human resources for mental health per 100,000 individuals are as low as 2.02 in the Philippines [2], and the mental health system still has many gaps between the needs of society and available resources. In such a situation of limited financial and human resources for mental health and psychosocial support, it is useful for the general population to learn how to provide mental health and psychosocial support among each other through PFA. PFA can be applied not only in times of disaster but also in daily life, including responding to stressful events such as the loss of significant others or sexual- and gender-based violence. 

Prior research has reported the effectiveness of face-to-face PFA orientation for nursing students [6], psychology students and school counselors [7], and practitioners from mental health welfare centers [8]. However, to the best of our knowledge, no studies have examined the effectiveness of PFA orientation for the general population since the PFA orientation tends to have been provided to those who have a role in providing support to others including health workers, public service providers, teachers, and volunteers, among others. 

Our research team previously developed an English version of the PFA e-orientation with the permission of the WHO. The e-orientation is composed of video material accessible through YouTube with pictures, narrations, and some exercises. The duration of the e-orientation is about two hours. All the modules are intended to be accessible to various populations by adding voice guidance and captioning. Its efficacy among health professionals was indicated in a study in Malaysia [9].

This study aimed to examine the effectiveness of the PFA e-orientation as well as face-to-face orientation among the general population in Muntinlupa City, the Philippines.

## 2. Methods

### 2.1. Participants and Intervention Procedures

Individuals over 18 years old in Muntinlupa City, the Philippines, were able to participate in this quasi-experimental study. Muntinlupa City is located on the island of Luzon and is part of Metro Manila. It has a population of 504,509 individuals as of 2015 [10]. The participants were 800 people randomly selected from 8 barangays (districts) as part of a health promotion event by the city government that was based on the official resident registry. The participants were only eligible if they were at least 18 years of age, and there were no other exclusion criteria including their literacy and disability. As part of recruitment for the study, participants were asked to participate spontaneously in health-related education. Data collection was conducted on 3 and 4 December 2019, at the building of Muntinlupa City Technical Institute. Among the 800 residents contacted by the city, 410 individuals participated in the present study. Each participant received 300 pesos as remuneration to join the study (about USD 6.24 as of 14 January 2021, according to Morningstar, Inc.), which is equivalent to half of the minimum daily wage in Metro Manila, the Philippines, according to the National Wages and Productivity Commission of the Philippines.

The participants were assigned to the following three groups: (1) the e-orientation group, (2) the face-to-face orientation (F2F) group, and (3) the control group, in the order of their arrival at the venue. The e-orientation group received a two-hour PFA e-orientation with audio and captioning in Filipino. The F2F group received a two-hour face-to-face PFA orientation on the same subject facilitated by a mental health professional/PFA instructor (Prof. Tsutsumi, Kanazawa University). The control group received a two-hour health promotion orientation for obesity prevention instructed by a primary health professional, in person as well (Prof. Umeda, University of Hyogo). This control group was established to consider whether possible differences between pre- and post-interventions are due to time and other factors. The F2F group and the control group orientation sessions were conducted in English with the interpretation of Filipino conducted by local students majoring in health sciences. All interventions were conducted with 30-40 participants per intervention.

This research was conducted in close partnership with the local government of Muntinlupa as part of a technical support initiative for strengthening the capacity of the mental health and psychosocial support system in the city.

### 2.2. Development of a Filipino Version of the PFA E-Orientation

The Filipino version of the PFA e-orientation was developed on the basis of the existing English version that our research team developed previously together with the National Institute of Mental Health, Japan; the UN University International Institute for Global Health; the UN Population Fund, and other partners. Regarding the translation procedure, first, translation from English to Filipino was conducted by a professional translation agency. Second, doctoral students from the Philippines at Kanazawa University reviewed the translated script. Third, an experienced government official working on mental health and psychosocial support in Muntinlupa City made necessary amendments in accordance with the local context. Finally, the amended script was back-translated into English by a different professional translator and assessed by bilingual health professionals. 

### 2.3. Questionnaire 

In this study, a structured questionnaire was employed. The questionnaire included the following: (1) socio-demographic data such as gender, age, years of education, and monthly-income, and (2) two checklists included in the PFA Facilitator’s Manual [11] for assessing the effectiveness of the PFA orientation. A checklist titled “Confidence to Provide PFA” (PFA Confidence) is an eight-item checklist aimed at evaluating the confidence to properly provide PFA following a disaster. These include “Ability to support people who have experienced disasters and other stressful events” and “Ability to listen in a supportive way”, among others. It contains a 5-point response set (i.e., very low, low, medium, high, very high) for each item, and the total score (range: 8–40) is calculated. The higher the total score, the more confident the respondent is in providing PFA appropriately. The other checklist titled “Knowledge of PFA” (PFA Knowledge) is utilized to assess the essential knowledge of PFA. This study used a section titled “Things which are helpful for people after a very distressing event” that contains 10 items. For example, questions such as “Asking people to recount their traumatic experiences in detail” and “Telling them the story of someone else you just saw so that they know they are not alone” were included. These items are dichotomous responses with a “Yes” or “No” option. The structured questionnaire was also translated using the same procedure as for the PFA e-orientation as described. Cronbach’s alpha for PFA Confidence was 0.86 at pre-intervention and 0.93 at post-intervention. Thus, sufficient internal consistency was confirmed, and hence the total sum score was used in the analysis. Cronbach’s alpha for PFA knowledge was 0.32 at pre-intervention and 0.48 at post-intervention. These demonstrated low internal consistency. Therefore, the difference in the number of correct answers between the pre- and post-interventions was analyzed for each question. The questionnaire booklet containing both pre- and post-intervention questions was distributed to each participant at the beginning of the study. The participants answered these checklists before and after the interventions.

### 2.4. Statistical Analysis

As a result of data cleaning, four participants were excluded: Three participants who reported their age was under 18 years old and one participant who reported 57 years of education, since we judged that the reliability of their answers could be compromised. This resulted in a total of 406 participants included in the final analysis: 150 participants in the e-orientation group, 139 participants in the F2F group, and 117 participants in the control group. Socio-demographic data were compared among three groups using a Fisher’s exact test for gender, analysis of variance (ANOVA) tests for age and education years, and a Kruskal–Wallis test for monthly income.

In the PFA confidence analysis, participants who did not answer two or more of the eight questions were excluded from the analysis, resulting in 392 participants in total: 145 participants in the e-orientation group, 134 participants in the F2F group, and 113 participants in the control group. For participants who did not answer one question, the mean of the other seven questions was substituted for the missing value. The total PFA Confidence scores at pre-intervention were compared using an ANOVA test. The comparison of the mean scores between the pre- and post-interventions for each group was analyzed using paired t-tests.

For each question in PFA Knowledge, the comparison of the number of correct answers between the pre- and post-interventions for each group was analyzed using a McNemar test. 

Statistical analysis was conducted using IBM SPSS Statistics 25. Statistical significance was set at *p* < 0.05.

### 2.5. Ethical Considerations

The participants were provided a verbal explanation of the study along with an explanatory written document of the research. All participants were requested to sign the consent form for final confirmation prior to joining the study. All of the obtained data and information was quantified and safely stored to secure personal information. All the procedures for this study were approved by the ethical committee of the Organization of Global Affairs, Kanazawa University (no. KINDAIKOKUKI-001GO).

## 3. Results

The socio-demographic characteristics of the three groups are presented in Table 1. The percentages of men, women, and others were 31.3%, 64.7%, and 4.0% in the e-orientation group; 29.5%, 68.3%, and 2.2% in the F2F group; and 41.0%, 55.6%, and 3.4% in the control group, respectively. The mean age was 33.4 (SD = 12.1) in the e-orientation group, 35.0 (SD = 13.8) in the F2F group, and 34.2 (SD = 13.8) in the control group. The mean years of education were 10.4 (SD = 3.5) in the e-orientation group, 10.9 (SD = 3.0) in the F2F group, and 11.4 (SD = 3.0) in the control group. The mean rank of monthly income was 169.0 in the e-orientation group, 191.4 in the F2F group, and 200.7 in the control group. There were no significant differences among the three groups in any of the demographic characteristics, while there were trends towards significance in education years and monthly income. 

In order to see the differences in confidence scores at pre-intervention among the three groups, an ANOVA was employed. As indicated in Figure 2, there was a significant difference in total scores at the pre-intervention period, and the score of the control group was significantly higher than that of the e-orientation group. Therefore, analyses of means between pre- and post-intervention scores were performed in each group by employing a paired *t*-test. Figure 2 shows the results of the comparison. In the e-orientation group, the mean score of pre-intervention was 25.1 (SD = 4.7), and the mean score of post-intervention was 26.1 (SD = 5.3). In the F2F group, the mean score of pre-intervention was 26.2 (SD = 6.0), and the mean score of post-intervention was 29.6 (SD = 6.9). In the control group, the mean score of pre-intervention was 26.6 (SD = 3.6), and the mean score of post-intervention was 27.2 (SD = 5.1). The results demonstrated that the mean scores of the e-orientation and F2F groups increased significantly, while the control group showed no significant difference. 

Of the 10 knowledge-related questions, those questions where more than 90% of respondents answered correctly (i.e., Questions 1, 3, and 9) at the pre-intervention were excluded from further analysis as it was difficult to assess the effect of the intervention. The remaining seven questions were analyzed. 

Table 2 shows the results comparing the number of correct answers between the pre- and post-interventions for each group using the McNemar test. In the e-orientation group, correct answers significantly increased in one question (i.e., Question 5), and there was a trend for improvement in another question (i.e., Question 2). In the F2F group, correct answers increased in two questions (i.e., Questions 6 and 7), and there were trends for improvement in two questions (i.e., Questions 2 and 5). In the control group, there were no significant differences in any of the questions between pre- and post-intervention times.

## 4. Discussion

This study examined the effectiveness of the PFA e-orientation and face-to-face orientation, in comparison with non-mental health orientation, among the general population in Muntinlupa City, the Philippines. The results of the comparison between pre- and post-interventions for each group suggested that both the PFA e-orientation and face-to-face orientation had positive effects on the confidence and knowledge related to PFA. 

Because there was a significant difference in the pre-intervention scores among the three groups, a comparison of the post-intervention scores among the three groups was not employed. Instead, the comparison between the pre- and post-interventions was employed for each group separately. It is of note that the PFA confidence score of the control group was significantly higher at the pre-intervention time. The group allocation was randomly conducted in order of arrival at the venue; the difference is interpreted as a random error. While we did not officially include the outcomes in the study, utilizing the pre-intervention scores as a covariate, the post-intervention scores, as well as the differences in the mean scores between the pre- and post-interventions among the three groups, were analyzed. The results showed that significant differences were detected regarding the post-intervention scores and the difference in scores between pre- and post-interventions for the F2F group only.

In addition, the knowledge of PFA increased significantly for both the PFA e-orientation and face-to-face orientation groups, while no significant differences were detected in the control group. When examining the content of the questions, the number of correct answers for the question stating “Not telling them the story of someone else you just saw so that they know they are not alone” increased significantly in the e-orientation group and showed a trend of increase in the F2F group. Additionally, in the F2F group, the number of correct answers to the questions stating “Not making promises so that people feel better” and “Not telling an affected person that everything will be fine and they should not worry” increased significantly. The number of correct answers to the question stating “Not asking people to recount their traumatic experiences in detail” showed trends of increase in both the e-orientation group and the F2F group. All these factors, such as breaking confidentiality, providing incorrect information, and recounting extremely stressful experiences in detail, can be harmful and may increase the probability of post-traumatic stress disorder (PTSD) and other deteriorations in mental health and psychosocial well-being later in life [3]. In conclusion, the results of this study indicate that both e-orientation and F2F orientation have important effects in preventing possible harm.

Overall, the results suggest that both the PFA e-orientation and face-to-face orientation are effective for the general population in terms of both increasing confidence and knowledge. This is the first study to show the effectiveness of the PFA orientation among those who are not specialists. This result, which indicates that e-orientation can be useful in increasing confidence and knowledge, is important since financial and human resources for mental health are limited in developing countries, and online interventions can readily increase both coverage and accessibility. This is especially true in the ongoing situation of the COVID-19 pandemic. 

One limitation of the study may be the lack of score growth. The reason for this may be that the orientation was only offered for about two hours to the general population, who are not necessarily interested or motivated in learning about mental health and psychosocial well-being and PFA. In particular, e-orientation was offered without much interaction. In the future, it may be beneficial to provide e-orientation by dividing the orientation into several sessions, repeating the orientation, or including interactive activities. 

A second limitation of this study was the possibility that the results were affected by several conditions. First, due to typhoons and traffic conditions, the number of people who could come to the study might have depended on the vicinity of the research site. In particular, some individuals with disabilities might have experienced greater barriers and could not participate because of accessibility issues as a result of the typhoon. Second, because the orientation was conducted during the daytime on weekdays, the percentage of women and those who were not employed were high. Third, those whose addresses were not listed in the official resident registry, such as individuals living in the slums, were not included. It is necessary to conduct further research in order to address these limitations and further look into effectiveness among various groups including marginalized populations who tend to be simultaneously susceptible to risks for mental health conditions.

E-learning is specifically gaining strong attention due to the COVID-19 pandemic worldwide. The outcome of this study might be able to be utilized in designing e-learning in the COVID-19 era for the general population. In addition, PFA e-orientation, which the general population can learn easily, may bring a significant contribution to protecting and promoting mental health and psychosocial well-being in the COVID-19 pandemic and beyond at the community level and increasing the resilience and mutual support of the community.

## 5. Conclusions

This study investigated the effectiveness of the PFA orientation among the general population and through e-orientation for the first time. The results suggested that both the PFA e-orientation and face-to-face orientation were effective for the general population in terms of both increasing confidence and knowledge related to PFA. The PFA orientation for the general public including the e-orientation might be able to contribute to increased access to basic mental health and psychosocial support in resource-constrained settings including developing countries, archipelagic nations, and in the COVID-19 pandemic.

## Figures and Tables

**Figure 1 ijerph-20-00983-f001:**
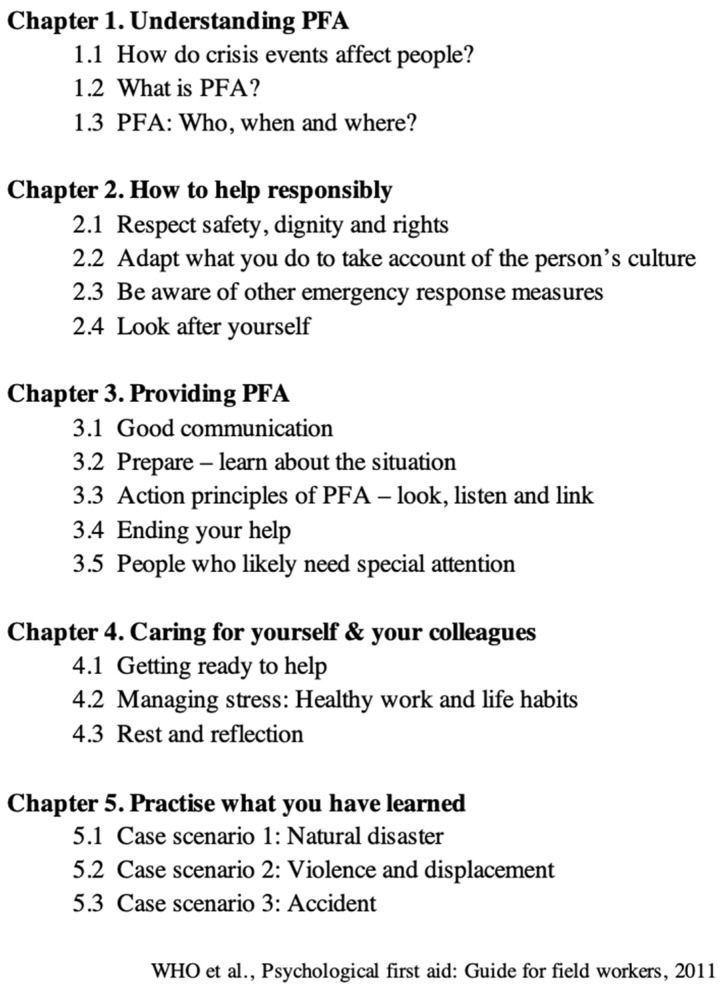
Psychological first aid: Guide for field workers table of contents [5].

**Figure 2 ijerph-20-00983-f002:**
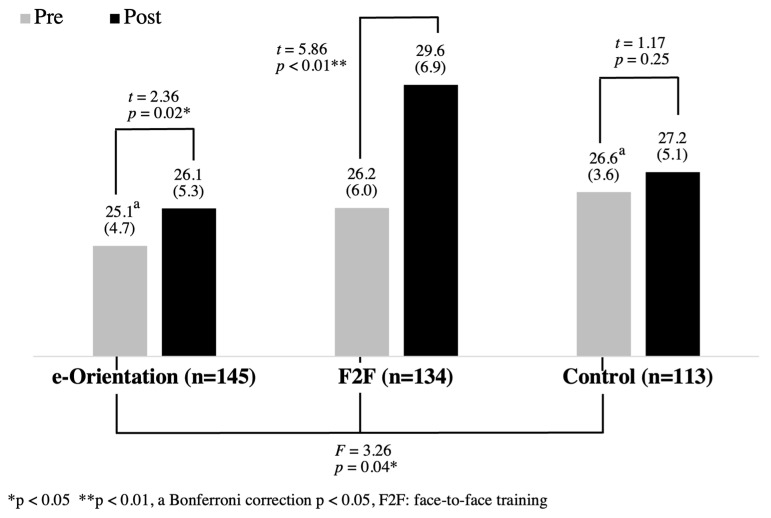
The PFA Confidence scores of the pre- and post-interventions (n = 392).

**Table 1 ijerph-20-00983-t001:** Socio-demographic characteristics (n = 406).

	e-Orientation (n = 150)	F2F (n = 139)	Control (n = 117)		
	n (%)	n (%)	n (%)		*p*
Gender					0.25 ^a^
Male	47 (31.3)	41 (29.5)	48 (41.0)		
Female	97 (64.7)	95 (68.3)	65 (55.6)		
Others	6 (4.0)	3 (2.2)	4 (3.4)		
	**Mean (SD)/Mean Rank**	**Mean (SD)/Mean Rank**	**Mean (SD)/Mean Rank**	* **F/H** *	* **p** *
Age	33.4 (12.1)	35.0 (13.8)	34.2 (13.8)	0.46	0.64
Education years	10.4 (3.49) ^b^	10.9 (3.03)	11.4 (3.03) ^b^	2.64	0.07 ^†^
Monthly income	169.0	191.4	200.7	5.93	0.05 ^†^

^†^ *p* < 0.1; ^a^, Fisher’s exact test; ^b^, Bonferroni correction *p* < 0.05; F2F, face-to-face training.

**Table 2 ijerph-20-00983-t002:** The comparison of the number of correct answers between the pre- and post-interventions (n = 406).

		e-Orientation (n = 150)				F2F (n = 139)				Control (n = 117)		
		n		*χ^2^*	*p*	n		*χ^2^*	*p*	n		*p* ^a^
		pre	post			pre	post			pre	post	
2	Not asking people to recount their traumatic experiences in detail	63	76	3.51	0.06 ^†^	69	82	3.69	0.06 ^†^	65	66	1.00
4	Not conducting psychological debriefing	28	33	0.59	0.44	31	38	1.09	0.30	21	20	1.00
5	Not telling them the story of someone else you just saw so that they know they are not alone	10	24	6.04	0.01 *	17	27	2.89	0.09 ^†^	10	14	0.48
6	Not making promises so that people feel better	21	25		0.54 ^a^	21	38	6.92	<0.01 **	27	23	0.48
7	Not telling an affected person that everything will be fine and they should not worry	8	15		0.19 ^a^	5	14		0.05 *^a^	5	9	0.29
8	Not judging the person’s actions and behavior so they won’t make the same mistakes next time	58	62	0.24	0.63	74	76	0.02	0.88	55	62	0.21
10	Not telling an affected person how they should be feeling	22	23	0.00	1.00	34	38	0.30	0.58	21	21	1.00

^†^*p* < 0.1; * *p* < 0.05; ** *p* < 0.01; ^a^, binomial distribution used; F2F: face-to-face training.

## Data Availability

Not applicable.

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
