# Peer review of "Effectiveness of Psychological First Aid e-Orientation among the General Population in Muntinlupa, the Philippines"

_ijerph, 2023, doi:10.3390/ijerph20020983_

Round 1

Reviewer 1 Report

Thank you for giving a chance to review the manuscript. There are some suggestions to improve the quality of manuscript. 

Abstract:

The method and result section of the Abstract is not clear. It should clearly tell what assessment methods used to see the effectiveness. Result section should be revised to clearly demonstrate what are main results in accordance with study objective. Should report significance level and mean scores in comparative analysis. “there was an upward trend in another question” this expression and language both are unclear and inappropriate for such kind of results.

Introduction

You have given some arguments for PFA by general population.  However, given the fact that you reported (However, to the best of our knowledge, no studies have examined the effectiveness of PFA orientation for the general population). (Please report some of the inherent challenges or difficulties in training of PFA with general population. What was the reluctance for not training general population in PFA earlier). Elaborate some point to justify the scope and significance of this study.

Method:

When describing sampling methodology please clarify. (The city government randomly selected 800 people) Have you applied random sampling procedures? If yes, then it involves listing all individuals in the city and then applying procedures for selection of participants or it was by convenience.  It seems it is convenient sampling so please revise the sampling methodology. Plus report some inclusion/exclusion criterion for general population because this training might require some education level/own health of people from general population/disability condition/ prior experience in such training/counseling experience etc.

The information related to reliability analysis of assessment tools should be reported where describing the tool rather in statistical analysis section.

Results:

The results are interesting, but one thing not making sense here that control group participants have higher score in pre-training condition. Is it due to statistical reasons or due to earlier exposure of control group participants in any such similar training or their prior education in field of psychology?

Discussion:

There is need to improve the expression and language in discussion section. Please also discuss how the elevated scores in pre-training for control group could be reason for non-significant difference in post-training.  Because the score is based correct and wrong answers, so if they have already made correct answers, post-assessment would have given same answers. See this aspect as well while discussing the findings.

Overall, there is need to discuss the implications of findings, any other factors that could have influenced the results and their interpretation.

There is need to improve the expression throughout the manuscript.  I have never heard a word such as ‘supplementally” mentioned in discussion (Line 232)

In line 249 the reference is given by different format, because all other references are numbered and not given in brackets.

Conclusion:

This suggests that the  PFA e-orientation could be useful in preventing harm in mental health and psychosocial  support as well as promoting resilience in resource constrained settings including developing countries, archipelagic nations, and in the COVID-19 pandemic, together with face- to-face orientation. (How this can be concluded when the mental health impacts were not assessed nor reported in this analysis.) This study only assessed the differences in training methods of PAF for e-orientation and face-to-face groups and with control group.

Author Response

We wish to express our appreciation to you for your insightful comments on our paper. The comments have helped us significantly improve the paper.

We have made changes to the entire paper to reflect your comments and would be happy to review it.

As for the results of the Results, we will respond below.

Thank you for your important question.
We suspect that the differences between the groups happened by chance. Since the subjects were randomly allocated to groups in the order of their arrival at the venue: 1st comer was allocated to e-Orientaion group, 2nd to F2F group, 3rd to ctrl group, 4th to e-Orientation, 5th to F2F… We added this in the discussion section.

Reviewer 2 Report

Comments to the authors

Manuscript's title: Effectiveness of Psychological First Aid e-Orientation among 

the General Population in Muntinlupa, Philippines

Thank you for the opportunity to review this manuscript regarding the First Aid e-Orientation in the Philippines. This is an important topic, the manuscript is written well, but I outline some needed clarifications.

Abstract

1.     Please add some information regarding the analysis method

2.     It Is mentioned in the abstract: "the control group consisted of 117 participants who received a two-hour health promotion orientation for obesity." how do they get the orientation in the control group?

Introduction

3.     The introduction is well-written. However, it could be shorter, and some information regarding the rates of mental health problems in the Philippines needs to be included.

4.     The rationale for having the control group needs to be understood. Please clarify it.

5.     Please be aware that there are no refs in some paragraphs. For example, on page 2, lines 62-69.

Methods

6.     Please clarify how the government randomly selected 800 people.

7.     Please provide an example item for each part of the questionnaire.

8.     How were the final scores calculated? Mean? Sum?

9.     Please provide detail about the ethical institute's permission.

Discussion

10.  The discussion does not include any integration with previous studies or current literature on the topic. Please re-write the discussion while taking this point into account.

11.  Please expand your explanation regarding theoretical and practical implications.

Author Response

We wish to express our appreciation to you for your insightful comments on our paper. The comments have helped us significantly improve the paper.

We have made changes to the entire paper to reflect your comments and would be happy to review it.
